# Single Amino Acid Substitutions in the Cucumber Mosaic Virus 1a Protein Induce Necrotic Cell Death in Virus-Inoculated Leaves without Affecting Virus Multiplication

**DOI:** 10.3390/v12010091

**Published:** 2020-01-13

**Authors:** Ainan Tian, Shuhei Miyashita, Sugihiro Ando, Hideki Takahashi

**Affiliations:** Graduate School of Agricultural Science, Tohoku University, 468-1, Aramaki-Aza-Aoba, Sendai 980-0845, Japan; imtenten@me.com (A.T.); shuhei27@gmail.com (S.M.); sugihiro.ando.a2@tohoku.ac.jp (S.A.)

**Keywords:** cell death, cucumber mosaic virus, hypersensitive response, methyltransferase domain, necrosis

## Abstract

When *Arabidopsis thaliana* ecotype Col-0 was inoculated with a series of reassortant viruses created by exchanging viral genomic RNAs between two strains of cucumber mosaic virus (CMV), CMV(Y), and CMV(H), cell death developed in the leaves inoculated with reassortant CMV carrying CMV(H) RNA1 encoding 1a protein, but not in noninoculated upper leaves. In general, cell death in virus-infected plants is a critical event for virus survival because virus multiplication is completely dependent on host cell metabolism. However, interestingly, this observed cell death did not affect either virus multiplication in the inoculated leaves or systemic spread to noninoculated upper leaves. Furthermore, the global gene expression pattern of the reassortant CMV-inoculated leaves undergoing cell death was clearly different from that in hypersensitive response (HR) cell death, which is coupled with resistance to CMV. These results indicated that the observed cell death does not appear to be HR cell death but rather necrotic cell death unrelated to CMV resistance. Interestingly, induction of this necrotic cell death depended on single amino acid substitutions in the N-terminal region surrounding the methyltransferase domain of the 1a protein. Thus, development of necrotic cell death might not be induced by non-specific damage as a result of virus multiplication, but by a virus protein-associated mechanism. The finding of CMV 1a protein-mediated induction of necrotic cell death in *A. thaliana*, which is not associated with virus resistance and HR cell death, has the potential to provide a new pathosystem to study the role of cell death in virus–host plant interactions.

## 1. Introduction

The role of cell death in virus–host plant interactions is a phenomenon that has long been discussed but has yet to be resolved [1,2,3,4,5]. Cell death in virus-infected plants is a critical event for the survival of the virus because virus multiplication is completely dependent on host cell metabolism. Cell death resulting from incompatible interactions between viruses and plants has been described as *necrotic local lesions*, and that occurring in compatible interactions as *necrotic cell death* [6,7]. Cell death observed as necrotic local lesions at primary viral infection sites on host plants that carry nucleotide-binding and leucine-rich repeat (NB-LRR) class R protein-coding virus resistance (*R*) genes, and infected with a virus carrying avirulence (*AVR*) gene encoding AVR protein, has been thoroughly analyzed: it has long been recognized as a hallmark of the hypersensitive response (HR) and R protein-mediated resistance to viruses [8,9,10,11]. Thus, cell death which develops at necrotic local lesions, is referred to as *HR cell death*. Also, it is now considered a form of programmed cell death (PCD) due to similarities in the cytological and physiological features of this kind of cell death between plants and animals, despite some substantial differences; for example, while plant PCD exhibits vacuolar death, animal PCD does not [11,12,13,14]. Thus, HR cell death has been well characterized. However, while HR cell death should be critical for virus multiplication, viruses are still able to move into the living cells surrounding the necrotic local lesions, and prevention of the further spread of viruses into living cells surrounding necrotic local lesions is observed [15,16,17,18]. Thus, the role of HR cell death in virus resistance is still unclear.

In comparison with HR cell death, necrotic cell death seems to be poorly understood, although a limited number of studies have been reported. For example, cucumber mosaic virus (CMV)-induced cell death was observed on inoculation of *A. thaliana* leaves with a lily strain of CMV [CMV(HL)], and it was concluded that this necrotic cell death was caused by reduction of host catalase activity through direct interaction between CMV(HL) 2b protein and catalase, thereby preventing production of scavenging cellular hydrogen peroxide and resulting in necrotic cell death [19]. However, it remains unclear if necrotic cell death resulted from non-specific damage to host cells caused by CMV(HL) infection, rather than as a form of programmed cell death. Other than the necrotic cell death that has been investigated, various types of necrotic cell death that are not well characterized seem to exist in various interactions between host plants and viruses [7].

CMV is one of the best characterized tripartite RNA viruses and has positive-sense single-stranded RNA genomes: RNA1, RNA2, and RNA3 [20]. RNA1 encodes the 1a protein, which has two putative functional domains: a methyltransferase (MET) domain in amino acid positions 72–290 and a helicase/NTP-binding (HEL) domain in amino acid positions 711–976 [21,22]. RNA2 encodes the 2a protein containing motifs of RNA-dependent RNA polymerase [22]. The 1a protein interacts with the 2a protein through the HEL domain in the yeast-two hybrid system [23] and these are thought to be components of a viral replicase complex [24]. RNA2 has a second open reading frame (ORF) encoding the 2b protein, which functions as a suppressor of post-transcriptional gene silencing (PTGS) [25,26,27,28]. RNA3 has two open reading frames: 3a and coat protein (CP) [29,30,31]. 3a encodes a cell-to-cell movement protein (3a protein). CP is translated from subgenomic RNA4, which is generated from the CP region of minus-stranded RNA3 in virus-infected cells. CMV has a large host range including *Arabidopsis thaliana* [20], and comparative and incompatible interactions between CMV strains and *A. thaliana* ecotypes have been well characterized at the molecular level [32].

Interestingly, in analysis of the host response to a series of reassortant viruses between two CMV strains with differing virulence in *Arabidopsis thaliana*, we discovered that cell death occurred in virus-inoculated leaves of *A. thaliana* ecotype Col-0 in response to a reassortant CMV. In the present study, this cell death phenomenon is characterized, and the viral determinant inducing cell death is identified. Several features of the cell death observed here indicated that it might not be HR cell death but rather necrotic cell death that does not affect CMV multiplication. Development of this necrotic cell death is determined by single amino acid residues in the N-terminal region surrounding the methyltransferase domain of the 1a protein encoded on CMV RNA1.

## 2. Materials and Methods

### 2.1. Plants and Virus

*Arabidopsis thaliana* ecotype Col-0 and other 94 ecotypes are listed in Appendix A. *RCY1*-transformed Col-0 (Col::pRCY1-HA#12) [33] which is renamed Col::RCY1 in the present study, and *Nicotiana benthamiana* were grown on soilless mix (Metro-Mix^®^ 380, Sun Gro Horticulture, Agawam, MA, USA) under a 14-h light (14,000 lux)/10-h dark photoperiod at 25 °C in a KG-201 HL-D growth chamber (Koito, Yokohama, Japan). Since *RCY1* was isolated from *A. thaliana* ecotype C24 as a NB-LRR class resistance gene to a yellow strain of CMV [CMV(Y)], Col::RCY1 was used as a control for developing HR cell death in response to CMV [33,34]. CMV(Y) [35] and the H strain of cucumber mosaic virus [CMV(H)], which was isolated from an *Arabidopsis halleri* plant showing no symptoms, were used for these experiments. Also used were a series of reassortant CMVs exchanging RNA1, 2, and 3 between CMV(Y) and CMV(H); CMV carrying chimeric RNA1 between CMV(Y) and CMV(H) and CMV(Y); and CMV carrying single amino acid substitutions in CMV(Y) 1a protein encoded in CMV RNA1.

### 2.2. In Vitro Transcription of Infectious CMV RNA and Production of Reassortant CMV

Infectious CMV(Y) RNA1, RNA2, and RNA3 were transcribed in vitro pCY1-T7, pCY2-T7, and pCY3-T7, respectively [36]. cDNAs of CMV(H) RNA1, 2, and 3 were synthesized by RT-PCR according to standard protocol [37], with the following sets of primers: CMV.RNA1-5’.F and CMV.RNA1-3’.R for RNA1; CMV.RNA2-5’.F and CMV.RNA2-3’.R for RNA2; and CMV.RNA3-5’.F and CMV.RNA3-3’.R for RNA3 (Appendix A). The sequence encoding T7 RNA polymerase was included in primers CMV.RNA1-5’.F, CMV.RNA2-5′.F, and CMV.RNA3-5’.F (Appendix A). RT-PCR reactions were performed using the PrimeScript™ II High Fidelity One Step RT-PCR Kit (Takara Bio, Shiga, Japan) according to the manufacturer’s instructions. All PCR products were purified with the Wizard^®^ SV Gel and PCR Clean-Up System (Promega, Madison, WI, USA). The gel-purified cDNA of RNA1 was cloned into the *Hin*dIII and *Not*I sites of pUC118 (Takara Bio) and the cDNAs of RNA2 and RNA3 were cloned into the *Bam*HI and *Not*I sites using the In-Fusion HD Cloning System (Takara Bio) according to the manufacturer’s instructions. The plasmid constructs containing each of the CMV(H) RNA1, RNA2, and RNA3 cDNAs (designated pCH1-T7, pCH2-T7, and pCH3-T7, respectively) were linearized by digestion with *Not*I and purified using the Wizard^®^ SV Gel and PCR Clean-Up System (Promega). Each linearized plasmid DNA was then transcribed in vitro using T7 RNA polymerase with the standard AmpliCap-Max™ T7 High Yield Message Maker Kit (Cellscript, Madison, WI, USA) according to the manufacturer’s instructions.

To generate the reassortant CMVs including CMV(HYY), CMV(YHY), CMV(YHH), CMV(YYH), CMV(HYH), and CMV(HHY) (Appendix A), each infectious CMV RNA1, RNA2, and RNA3 was reciprocally exchanged between CMV(Y) and CMV(H). Four-week-old *N. benthamiana* was rub-inoculated with a combination of infectious CMV(Y) and CMV(H) RNA1, RNA2, and RNA3 to propagate a series of reassortant CMVs. At 7 days post-inoculation (dpi), the inoculated leaves were collected and weighed, and then ground in 10× volume of 0.1 M phosphate-buffered saline (pH 8.0) on ice. These homogenates were used to inoculate new fully expanded leaves of 6-week-old *N. benthamiana* plants. At 4 dpi, the inoculated leaves were harvested and used for virus purification. Virus purification was performed according to a previously described procedure [38]. 

### 2.3. Virus Inoculation and Detection

Fully expanded leaves of *A. thaliana* were rub-inoculated with 100 μg/mL of virus as previously described [39]. Virus was detected immunologically by western blot analysis according to the standard protocol [37] using antibody against the CP of CMV.

Accumulation of CMV RNA in virus-inoculated leaves of Col-0 was analyzed by northern hybridization according to the standard protocol [37]. CMV RNA-specific cDNA probes complementary to the 3′ noncoding region of all CMV RNAs were amplified from CMV(Y) RNA3 cDNA with a pair of primers: 5′-GTGAACGGGTTGTCCATCCA-3′ and 5′-ACCCTGAAACTAGCACGTTGT-3′ by PCR. The probe cDNA was labeled with digoxigenin (DIG)-11-dUTP using a DIG PCR labeling kit (Roche, Penzberg, Germany) according to the manufacturer’s instructions. The PCR product was purified according to the procedure of Takahashi and Ehara [38]. Ribosomal RNA (rRNA)-specific probe was obtained as described previously [33]. All CMV RNAs were detected using an alkaline phosphatase conjugated anti-DIG antibody (Roche, Penzberg, Germany) and visualized with the CDP-Star Reagent (New England Biolabs, Beverly, MA, USA) according to the manufacturer’s protocols.

After *A. thaliana* ecotype Col-0 was inoculated with CMV containing chimeric RNA1 or RNA1 carrying a nucleotide substitution, all of the chimeric RNA1 cDNAs and single nucleotide substitution RNA1 cDNAs were amplified by RT-PCR from the upper noninoculated leaves of inoculated plants. RT-PCR-amplified fragments were purified by treatment with ExoSAP-IT PCR Clean Up Reagents (Thermo Fisher Scientific, Waltham, MA, USA) according to the instruction manual, and their nucleotide sequences were confirmed by Sanger sequencing using a CEQ8000 Automated DNA Sequencer (Beckman Coulter, Brea, CA, USA).

### 2.4. Virus Quantification by ELISA

For quantitative measurement of the CMV CP by ELISA, three independent virus-inoculated leaves were homogenized in a 10× volume of 0.01 M potassium phosphate buffer (pH 8.0). The protein concentrations of the homogenates were determined using the Bradford reagent [39]. The homogenates used for ELISA were adjusted to 0.03 mg/mL total protein with 0.01 M potassium phosphate buffer. CP quantities were measured using the method of Koenig (1981) [40] and expressed as the absorbance at 405 nm per 0.03 mg/mL of total protein. Statistical analysis of CP quantities were performed using one-way analysis of variance (ANOVA) and Fisher’s least significant difference LSD test for post-hoc comparisons using IBM SPSS Statistics version 25 (IBM, Armonk, NY, USA).

### 2.5. Detection of Cell Death

Cell death in CMV-inoculated leaves was visualized by staining with trypan blue according to a standard protocol [41]. Virus-inoculated leaves were stained by boiling for 8 min in alcoholic lactophenol [99.5% ethanol:phenol:glycerol:lactic acid 4:1:1:1 (*v*:*v*:*v*:*v*)] containing 0.1 mg/mL trypan blue. The stained leaves were decolorized in a 2.5 g/mL chloral hydrate solution overnight, and then held and pictured in 70% ethanol. Trypan blue staining is available to detect cell death qualitatively, but it has limitations in attempting to show a quantitative measure of cell death.

### 2.6. RNA-Seq Analysis

Three independent mock- and CMV(HYY)-inoculated Col-0 leaves showing necrotic cell death at 5 dpi and mock- and CMV(Y)-inoculated Col::RCY1 leaves showing HR cell death at 3 dpi were used for extraction of total RNAs with the RNeasy Plant Mini Kit (Qiagen GmbH, Hilden, Germany). cDNA libraries were prepared using the TruSeq Stranded Total RNA with Ribo-Zero Plant Kit (Illumina, San Diego, CA, USA) according to the manufacturer’s instructions. Approximately 2.7 − 3.9 × 10^5^ paired-end reads (75-bp × 2) were obtained for each sample using the Illumina MiSeq (Illumina). The raw sequence data were submitted to the NCBI Gene Expression Omnibus under accession number GSE137625. The sequence reads were processed using Trimmomatic version 0.38 (Am Mühlenberg, Altenau, Germany) [42] for adaptor trimming and quality filtering. The processed reads were mapped to the genome sequences of *A. thaliana* ecotype Col-0, CMV(Y), and CMV(H) using STAR version 2.7 (New York, NY, USA) [43] at default settings. Read counts per *A. thaliana* gene were retrieved using the quantification option in STAR, and were normalized and statistically tested using DESeq2 R package 3.7 (Boston, MA, USA) [44]. Adjusted *p*-values were calculated [45], and the threshold-adjusted *p*-value was set to 0.05 for the present study. Independent filtering in DESeq2 with an automatically optimized threshold was performed to filter out the genes with low mean normalized counts. Genes that passed independent filtering in both the necrotic cell death versus mock and HR cell death versus mock comparisons were further analyzed for differential expression. Genes with fold-change >4 or <0.25 at an adjusted *p*-value of <0.05 were considered to be differentially expressed genes (DEGs). DEGs with increased expression unique to HR cell death were classified as Class I; DEGs with commonly increased expression in HR cell death and necrotic cell death were classified as Class II; and DEGs with increased expression unique to necrotic cell death were classified as Class III. DEGs with decreased expression unique to HR cell death were classified as Class IV; DEGs with commonly decreased expression in HR cell death and necrotic cell death were classified as Class V; and DEGs with decreased expression unique to necrotic cell death were classified as Class VI. The VennDiagram package (Toronto, Ontario, Canada) [46] was used to generate Venn diagrams of the sets of DEGs that overlapped between HR cell death and necrotic cell death. Gene symbols and gene ontology (GO) information were extracted using Metascape (http://metascape.org/gp/index.html) (accessed on 11th, January, 2020) [47]. GO enrichment analysis was implemented using ClusterProfiler package 3.14.0 (Guangzhou, Guangdong, China) in R software for the DEGs in each class [48].

### 2.7. Construction of In Vitro Transcription Vectors Carrying Chimeric cDNA of CMV RNA1

In vitro transcription vectors carrying chimeric forms of the region encoding the 1a protein in the RNA1 cDNA of CMV(H) or CMV(Y) were constructed as described below to generate vectors Y-H/683, H-Y/683 (Figure 7A); Y-H/343 and Y-H/344~682 (Figure 8A); and then vectors Y-H/71, Y-H/72~343, Y-H/290, Y-H/72~290, Y-H/71 + 291~343, and Y-H/291~343 (Figure 9A). All chimeric forms of the RNA1 cDNA region encoding 1a protein were generated by two-step PCR. First, the 3′- and 5’-fragments of RNA1 cDNA were amplified using CMV(H) or CMV(Y) cDNA as a template with the primers CMV RNA1-5’.FOR (Appendix A) and an internal reverse primer based on the reverse-strand sequence of the junction site for chimeric constructs (Appendix A). Secondary PCR products were also amplified using an additional internal forward primer complementary to the reverse primer (Appendix A) and the CMV.RNA1-3’.REV primer (Appendix A) used in the primary PCR. In some instances, such as for vectors Y-H/71, Y-H/72~343, Y-H/290, Y-H/72~290, Y-H/71 + 291~343, and Y-H/291~343 (Figure 9A), tertiary internal fragments were also amplified by RT-PCR using another set of internal primers. The sets of internal PCR primers used are listed in Appendix A.

In the second round of PCR, the resulting 5′ and 3′ fragments of RNA1 cDNA (and a third internal RT-PCR fragment, when necessary) amplified in the first PCR were used as templates to produce full-length chimeric RNA1 cDNA by PCR using the primers CMV.RNA1-5’.F and CMV.RNA1-3’.R (Appendix A). All PCR products were purified using the Wizard^®^ SV Gel and PCR Clean-Up System (Promega). The gel-purified RNA1 cDNA fragment was cloned into the *Hin*dIII and *Not*I sites of pUC118 (Takara Bio) with the In-Fusion HD Cloning System (Takara Bio) according to the manufacturer’s instructions. The nucleotide sequences of vector constructs carrying chimeric CMV RNA1 cDNAs were confirmed by Sanger sequencing using a CEQ8000 Automated DNA Sequencer (Beckman Coulter).

### 2.8. Single Amino Acid Substitution in 1a Proteins Encoded on RNA1 of CMV(Y)

Amino acid substitutions in the 1a protein were performed by generating site-directed mutant cDNAs by nucleotide substitution in pCY1-T7 using the GENEART Site-Directed Mutagenesis System (Thermo Fisher Scientific) with primers designed according to the manufacturer’s instructions. The primers used for nucleotide substitution are shown in Appendix A. All constructs were confirmed by Sanger sequencing using a CEQ8000 Automated DNA Sequencer (Beckman Coulter). In vitro transcription vectors carrying nucleotide substitutions in the 1a protein-coding region of the CMV(Y) RNA1 cDNA were designated T29A, I49V, G54S, R298Q, G299R, and H310N (Figure 10A).

## 3. Results

### 3.1. Response of Arabidopsis thaliana Ecotype Col-0 to a Series of Reassortant CMVs

Schematic structures of the reassortant CMV RNA genomes of the two parent CMV strains, CMV(Y) and CMV(H), are shown in Appendix A. When fully-expanded leaves of three independent *A. thaliana* ecotype Col-0 plant were inoculated with one of the reassortant CMVs [CMV(HHY), CMV(HYY), CMV(YHH), CMV(YYH), CMV(YHY) or CMV(HYH); and CMV(Y) or CMV(H) strain as a control], a cell death developed at 5 dpi in those inoculated with three of the reassortant CMVs containing CMV(H) RNA1: CMV(HHY), CMV(HYY), or CMV(HYH). However, cell death did not occur in Col-0 leaves inoculated with other reassortant CMV, CMV(Y), or CMV(H) (Figure 1A and Appendix A). At 5 dpi, cell death (which affected a much larger area in comparison with HR cell death) developed in Col-0 leaves inoculated with CMV(HHY), CMV(HYY), or CMV(HYH); whereas HR cell death developed in CMV(Y)-inoculated leaves of *RCY1*-transformed Col-0 (Col::RCY1) at 3 dpi (Figure 1A and Appendix A). These results suggest that CMV(H) RNA1 might be associated with cell death development in reassortant CMV-inoculated leaves through its interaction with CMV(Y) RNA2 or CMV(Y) RNA3 and a characteristic of this cell death is to spread to a broader area around the virus primary infection site than occurs with HR cell death.

The intensities of CMV CP bands detected by western blot analysis in CMV(HYY)-, CMV(HHY)-, or CMV(HYH)-inoculated leaves of Col-0 exhibiting cell death were comparable to those in CMV(Y)-, CMV(H)-, or other reassortant CMV-inoculated Col-0 leaves showing no cell death at 5 dpi (Figure 1B). The accumulated level of CP in CMV(HYY)-inoculated leaves was also quantitatively similar to that in CMV(Y)-inoculated Col-0 showing no cell death, but significantly higher than in CMV(Y)-inoculated Col::RCY1 leaves showing HR cell death (Figure 1C and Appendix A). Furthermore, comparison of the intensity of the norther blot analysis bands of CMV RNA1, RNA2, and RNA3 among the leaves inoculated with eight CMVs [CMV(H), CMV(Y), or one of six reassortant CMVs] suggests that there is no significant correlation between the induction of cell death and the accumulated level of CMV RNAs or the ratio of CMV RNA1, RNA2, and RNA3 (Figure 2). These results indicate that the cell death developing on the leaves inoculated with reassortant CMV carrying CMV(H) RNA1, seems to not suppress virus replication but instead allows it to multiply at the same level as with a susceptible interaction.

To investigate whether or not the cell death developing on the leaves inoculated with CMV(HYY) carrying CMV(H) RNA1 affects virus systemic spread to noninoculated upper leaves, CP in noninoculated upper leaves of CMV(HYY)-infected Col-0, CMV(Y)-infected Col::RCY1, or mock-inoculated Col-0 was detected by western blot analysis (Figure 3A). CP only accumulated in noninoculated upper leaves of CMV(HYY)-inoculated plants showing systemic stunting and weak yellowing symptoms, but not in upper leaves of CMV(Y)-infected Col::RCY1 or mock-inoculated Col-0. Moreover, systemic cell death was not observed in CMV(HYY)-inoculated plants or CMV(Y)-infected Col::RCY1 or mock-inoculated Col-0 (Figure 3B). Thus, the cell death developing on the leaves inoculated with reassortant CMV carrying CMV(H) RNA1 seems to not contribute to the resistance to CMV, and therefore differs from HR cell death.

### 3.2. Response of *A. thaliana* Ecotypes to CMV(HYY)

To determine whether the cell death induced in CMV(HYY)-inoculated leaves of *A. thaliana* ecotype Col-0 is a general response in *A. thaliana* ecotypes, 94 ecotypes of *A. thaliana* were inoculated with CMV(HYY). Cell death developed in virus-inoculated leaves in 92 out of 94 ecotypes from 5 to 9 dpi, but not in the ecotypes Mt-0 and Stw-0 at 14 dpi (Figure 4A,B, and Appendix A). According to our repetitive experiments, we could not detect necrosis induction in CMV(HYY)-inoculated leaves of Stw-0 and Mt-0, even if we cultivated them more than one month after inoculation (data not shown). CMV CP was detected in CMV(HYY)-inoculated Mt-0 and Stw-0 at similar levels to Col-0, but not in the virus-inoculated leaves of the other 92 ecotypes (Figure 4C). Cell death was not observed on the upper leaves of any of the 94 ecotypes systemically infected with CMV(HYY) (data not shown). Thus, *A. thaliana* ecotypes appear to generally develop cell death in response to CMV(HYY).

### 3.3. Comparison of Global Gene Expression Pattern between Two Types of Cell Death in Arabidopsis Leaves

To further characterize cell death in CMV-inoculated Col-0 leaves, global gene expression patterns were compared by RNA-Seq analysis between CMV(HYY)-inoculated Col-0 leaves showing cell death and CMV(Y)-inoculated Col::RCY1 leaves showing characteristic HR cell death. Changes in transcript abundances in CMV(HYY)-inoculated Col-0 leaves showing cell death and CMV(Y)-inoculated Col::RCY1 showing HR cell death were compared against mock treatment controls for 5906 genes with sufficient read counts for statistical analyses (adjusted *p*-value < 0.05). Genes were considered DEGs (differentially expressed genes) during the analysis of DESeq2 (Appendix A) for a >4-fold increase in expression or a <0.25-fold decrease in expression at an adjusted *p*-value of <0.05. As shown in Figure 5, a total of 202 genes showed a >4-fold increase in transcript abundance. Of these, 35 transcripts (Class II) showed significant increase in common with CMV(HYY)-induced cell death and CMV(Y)-induced HR cell death; while 149 transcripts (Class I) showed a significant unique increase in CMV(Y)-induced HR; and 18 transcripts (Class III) showed a similar such increase in CMV(HYY)-induced cell death (Figure 5; Appendix A). Simultaneously, 62 genes showed a significant <0.25-fold decrease in transcript abundance (Figure 5). Of these, seven (Class V) showed a common significant decrease; 43 (Class IV) specifically decreased in CMV(Y)-induced HR; and 12 (Class VI) showed a decrease in CMV(HYY)-induced cell death (Figure 5; Appendix A). These results indicate that global gene expression patterns in CMV(HYY)-inoculated Col-0 leaves developing cell death is different from that in characteristic HR cell death coupled to *RCY1*-conferred CMV(Y) resistance.

The number of genes for which transcript expression increased >4-fold or decreased <0.25-fold in CMV(HYY)-inoculated Col-0 leaves showing cell death was much lower than that in CMV(Y)-inoculated Col::RCY1 leaves showing HR cell death (Figure 5). Furthermore, the genes (Class I genes) with increased transcript abundance in CMV(Y)-inoculated Col::RCY1 leaves encoded several defense-related proteins: chitinase, pathogenesis-related (PR) proteins, and WRKY transcription factors; several leucine-rich repeat kinases and receptor-like proteins; calcium-binding EF-hand family proteins; and glutathione *S*-transferases (Appendix A). Gene ontology (GO) enrichment analysis suggested that the top three GO enrichment term in the biology processes for the identified DEGs with increased expression specific to HR cell death were enriched in “systemic acquired resistance”, “salicylic acid (SA) metabolic process”, and “SA biosynthetic process” (Appendix A, Appendix A). Transcripts encoded by overlapping sets of genes (Class II genes) were enriched in GO terms “systemic acquired resistance”, “SA biosynthetic process”, and “cell death” and could therefore be associated generally with the induction of cell death (Appendix A, Appendix A). In contrast, transcripts of Class III genes specific to cell death in CMV(HYY)-inoculated leaves were enriched in GO terms such as “jasmonic acid (JA)-mediated signaling pathway”, “cellular response to JA stimulus”, and “response to JA” (Appendix A, Appendix A), perhaps indicating a form of necrotic cell death that does not contribute to the resistance to CMV.

### 3.4. Analysis of the Viral Sequence in CMV RNA1 Inducing Necrotic Cell Death in Virus-Inoculated Leaves

Induction of necrotic cell death in CMV(HYY)-inoculated Col-0 leaves but not in CMV(Y)-inoculated Col-0 leaves suggested that CMV(H) RNA1 is responsible for inducing necrotic cell death with co-infection of CMV(Y) RNA2 and RNA3 in virus-inoculated leaves (Figure 1). There are 26 non-synonymous amino acid substitutions in 1a protein encoded on RNA1 of CMV(H) and CMV(Y) (Figure 6). To identify the region of 1a protein encoded by CMV(H) RNA1 responsible for inducing necrotic cell death in virus-inoculated Col-0 leaves, a series of chimeric cDNAs between CMV(Y) and CMV(H) RNA1 were generated and cloned under the control of the T7 promoter (Figure 7A, Figure 8A and Figure 9A). Each infectious RNA1 was transcribed in vitro from each chimeric cDNA vector, combined with infectious RNA 2 and RNA3 from CMV(Y), and used as inoculum.

When fully expanded leaves of Col-0 were inoculated with CMV(Y-H/683) and CMV(H-Y/683), which contain chimeric regions from nucleotide positions 1144 to the 3′-end of the 1a protein-coding sequence (due to reciprocal RNA1 cDNA fragment exchanges between RNA1 cDNA vectors Y and H; Figure 7A), necrotic cell death developed in fully expanded Col-0 leaves inoculated with CMV(H-Y/683) containing infectious RNA1 transcribed from chimeric RNA1 cDNA vector H-Y/683, as well as the leaves inoculated with CMV(HYY) (Figure 7B). However, cell death did not occur in leaves inoculated with either CMV(Y-H/683) or CMV(Y) (Figure 7B). Systemic cell death was not observed in any upper leaves of either CMV(H-Y/683)- or CMV(Y-H/683)-inoculated plants (data not shown), although the CMV CP was detected in the upper leaves in similar amounts in CMV(H-Y/683) and CMV(Y-H/683) (Appendix A). These results suggest that the region encoding 1a protein of CMV(H), which does not contain the helicase (HEL) domain, is necessary to develop necrotic cell death.

When fully expanded leaves of Col-0 were inoculated with CMV(Y-H/343) and CMV(Y-H/344~682), which contain chimeric regions from nucleotide positions 1127 to 2143 in the 1a protein-coding sequence due to reciprocal RNA1 cDNA fragment exchanges between RNA1 cDNA vectors Y and H-Y/683 (Figure 8A), necrotic cell death developed on the leaves inoculated with CMV(Y-H/343), but not on those inoculated with CMV(Y-H/344~682) (Figure 8B). Systemic cell death was not observed in the upper leaves of CMV(Y-H/344~682)- or CMV(Y-H/343)-inoculated plants (data not shown), although the CMV CP was detected at similar levels both in virus-inoculated leaves and the noninoculated upper leaves of these plants compared with CMV(Y)- and CMV(HYY)-inoculated plants as controls (Appendix A). Thus, the determinant for inducing necrotic cell death seems to be located in the 5’ region of RNA1, which corresponds to nucleotide positions 1–1126 in the 1a protein-coding region and includes the methyltransferase (MET) domain.

A total of 11 amino acid substitutions remain in 1a protein encoding the region between nucleotide positions 1 and 1126 of RNA1 of CMV(H) and CMV(Y) (Figure 6). Next, to further delimit the determinant that induces necrotic cell death, fully expanded leaves of Col-0 were inoculated with CMV(Y-H/71), CMV(Y-H/72~343), CMV(Y-H/290), CMV(Y-H/72~290), CMV(Y-H/71 + 291~343), or CMV(Y-H/291~343), which carries chimeric regions (Figure 9A) from nucleotide positions 1 to 310, 311 to 967, and 968 to 1126 in their 1a protein coding regions, respectively (Figure 6 and Figure 9A). Necrotic cell death developed on Col-0 leaves inoculated with CMV(Y-H/71), CMV(Y-H/72~343), CMV(Y-H/290), CMV(Y-H/71 + 291~343), or CMV(Y-H/291~343), but not on those inoculated with CMV(Y-H/72~290) (Figure 9B). Systemic cell death was not observed in upper leaves of plants systemically infected with CMV(Y-H/71), CMV(Y-H/72~343), CMV(Y-H/290), CMV(Y-H/72~290), CMV(Y-H/71 + 291~343), or CMV(Y-H/291~343) (data not shown), although the CMV CP was detected in virus-inoculated leaves and noninoculated upper leaves at similar levels (Appendix A). Thus, the development of the necrotic cell death seems to be determined by two independent regions of the 1a protein-coding region from nucleotide positions 1–310 or 968–1126, which do not include the MET domain (Figure 6 and Figure 9A). These two distinct 1a protein-coding regions, which equally determine the induction of the necrotic cell death, contained three amino acid differences between CMV(Y) and CMV(H) (Figure 6 and Figure 9A), respectively.

### 3.5. Analysis of Single Amino Acid Substitutions in the CMV 1a Protein for Induction of Necrotic Cell Death in Virus-Inoculated Leaves

To determine which amino acid in each region of the 1a protein induces necrotic cell death, nucleotide substitutions resulting in single amino acid substitutions were generated in each CMV(Y) 1a protein-coding region (Figure 10A). Necrotic cell death was induced in Col-0 leaves inoculated with CMV(T29A), CMV(I49V), CMV(G54S), CMV(R298Q), CMV(G299R), and CMV(H310N), respectively (Figure 10B). Systemic cell death was not observed in upper leaves of plants systemically infected with CMV(T29A), CMV(I49V), CMV(G54S), CMV(R298Q), CMV(G299R), and CMV(H310N) (data not shown), although the CMV CP was detected in virus-inoculated leaves and noninoculated upper leaves at similar levels (Appendix A). These results indicate that amino acid residues 29, 49, 54, 298, 299, and 310, which are located around the MET domain in the 1a protein encoded on CMV(Y) RNA1, independently determine the induction of necrotic cell death upon co-infection of Col-0 leaves with CMV(Y) RNA2 and RNA3.

## 4. Discussion

The response of *A. thaliana* ecotype Col-0 to a series of reassortant viruses with different chimeric composition of two CMV strains of differing virulence, CMV(H) and CMV(Y), was that necrotic cell death developed only in virus-inoculated leaves of Col-0 infected with reassortant CMVs carrying CMV(H) RNA1; for example, CMV(HYY) as shown in Figure 1. The amount of CMV CP accumulated in CMV(HYY)-inoculated Col-0 leaves was similar to that in CMV(Y)-inoculated Col-0 leaves showing necrotic cell death was similar to that in CMV(Y)-inoculated Col-0 leaves, which are susceptible to CMV(Y) but show no cell death (Figure 1 and Appendix A). Furthermore, both CMV(HYY) and CMV(Y) spread systemically in Col-0 plants (Figure 3). Therefore, the necrotic cell death developing on leaves inoculated with reassortant CMVs carrying CMV(H) RNA1 does not confer resistance to CMV, which is different from what happens with HR cell death.

When assessing host responses to reassortant CMVs, it was first considered that necrotic cell death might be an artifact caused by a heterogenous interaction between the CMV(H) 1a protein and other proteins encoded on CMV(Y) RNA2 and CMV(Y) RNA3, because CMV(H) itself did not induce necrotic cell death in virus-inoculated Col-0 leaves (Figure 1A). However, the results from single amino acid substitutions in the CMV(Y) 1a protein, were found to determine the occurrence of necrotic cell death, indicating that the mechanism of this necrotic cell death is likely to be more complex than that considered initially. The development of necrotic cell death in virus-inoculated leaves of Col-0 was induced by co-infection of CMV(Y) RNA2 and CMV(Y) RNA3 with a CMV(Y) RNA1 encoding a 1a protein carrying single amino acid substitutions around its MET domain, and this cell death did not affect virus multiplication. So far, it has been reported that a single amino acid substitution from R to C at amino acid position 461 of CMV 1a protein produces an HR-like necrotic phenotype in virus-inoculated leaves of *Nicotiana tabacum*, although the substitution does not affect virus multiplication [49]. Modeling of 1a protein has also demonstrated structural changes in the 1a protein caused by amino acid substitutions at position 461, which is relevant to an HR-like cell death phenotype [50]. Also, the HEL domain of the 1a protein of CMV isolate P0 [CMV(P0)] determines *Cmr1*-conferred resistance to CMV in pepper in a gene-for-gene manner [51]. These findings suggests that necrotic cell death (rather than being an artifact of a heterogenous interaction between the CMV(H) 1a protein and other proteins encoded on CMV(Y) RNA2 or CMV(Y) RNA3), may be induced by single amino acid mutation occurring naturally in CMV(Y) 1a protein during CMV(Y) multiplication in host cells.

In the experiments reported here, single amino acid substitutions at residues 29, 49, 54, 298, 299, or 310 in both the N- and C-terminal regions around the MET domain (amino acid 72 to 290) of the 1a protein (Figure 6 and Figure 10) independently induced necrotic cell death in Col-0. Although mutation affecting the amino acid composition of the MET domain of 1a protein disrupt capping activities and virus replication [52], single amino acid substitutions at the N- and C-terminal regions around the MET domain did not affect virus multiplication and systemic spread in the host plants (Figure 10). It is also observed that the N-terminal region of the hinge located between the MET and HEL domains of the 1a protein appears to self-interact to form homodimers in a yeast two-hybrid system [23]. Thus, change in the degree of self-interaction or conformational modification of the homodimer structure of the 1a protein (which could resulted from single amino acid substitutions around its MET domain) might be associated with the induction of necrotic cell death in Col-0 leaves in response to CMV(Y) carrying single amino acid substitutions around the MET domain. Further study is necessary to elucidate the mechanisms by which necrotic cell death is induced by single amino acid substitutions around the MET domain. However, the results reported here suggest that necrotic cell death can occur without preventing virus infection and is not caused by the stress of virus infection but by specific interactions between the virus and its host plant.

In this study, the induction of necrotic cell death was confirmed among 92 ecotypes of *A. thaliana* in response to the infection of CMV(HYY), Stw-0 and Mt-0 abolished the induction of necrotic cell death. There may be underlying plant factors that confer the induction of necrotic cell death through its interaction with CMV(HYY), and they might be inactivated by mutation or disrupted by deletion in Stw-0 and Mt-0.

Recently, evidence is accumulating that systemic necrosis (which was considered as a symptom of compatible interaction between a virus and its host plant) may result from the induction of HR cell death with incomplete restriction of virus spread in host plants [53,54,55,56,57,58,59]. The lethal systemic cell death might have been caused by delayed HR cell death and escape of the virus to distant tissues, thereby leading to runaway cell death, because virus-induced lethal systemic necrosis is correlated with activation of defense-related signaling pathways (e.g., MAP-kinase cascade) which are associated with incompatible interactions between host plants carrying *R* genes and avirulent strains of virus [55,58]. However, necrotic cell death in CMV(HYY)-inoculated Col-0 leaves did not cause lethal necrosis, even though the virus particles did systemically spread to noninoculated upper leaves of Col-0 plants (Figure 3B). This finding indicates that the necrotic cell death that developed in CMV(HYY)-inoculated Col-0 leaves might be a symptom of a compatible interaction between *A. thaliana* Col-0 and CMV, but not a resistance response to CMV or lethal systemic cell death caused by delayed HR cell death. Indeed, comparative RNA-Seq analysis of CMV(HYY)-inoculated Col-0 leaves and CMV(Y)-inoculated Col::RCY1 leaves, indicated that the numbers of up- or downregulated genes in CMV(Y)-inoculated Col::RCY1 leaves showing HR cell death were much greater than those in CMV(HYY)-inoculated Col-0 leaves showing necrotic cell death.

Many genes whose expression was specifically upregulated in CMV(Y)-inoculated Col::RCY1 leaves encoded several SA signaling-dependent defense-related proteins, which might be associated with *RCY1*-conferred resistance to CMV through restricting virus spread around primary infection sites (Figure 5). However, a number of JA signaling-related genes (which seem not to affect virus multiplication) were upregulated in CMV(HYY)-inoculated Col-0 leaves (Figure 5). Therefore, the necrotic cell death observed in CMV(HYY)-inoculated Col-0 leaves might not be contributing to CMV resistance. Moreover, only a limited number of overlapping DEGs (which might be relevant to necrotic cell death and HR cell death) was identified (Figure 5). Further investigation of the function of gene products associated with the induction of cell death should reveal the significance of cell death in the interactions between viruses and host plants.

Cell death in virus-infected plants is a critical event for virus survival, because virus multiplication depends on host cell metabolism. However, the role of cell death in virus–host plant interactions remains poorly understood. The finding of necrotic cell death in *A. thaliana*, which is determined by CMV-encoded 1a protein but unrelated to CMV resistance and HR cell death (including lethal systemic cell death), presents a new pathosystem to investigate the role of necrotic cell death in virus–host plant interactions.

## Figures and Tables

**Figure 1 viruses-12-00091-f001:**
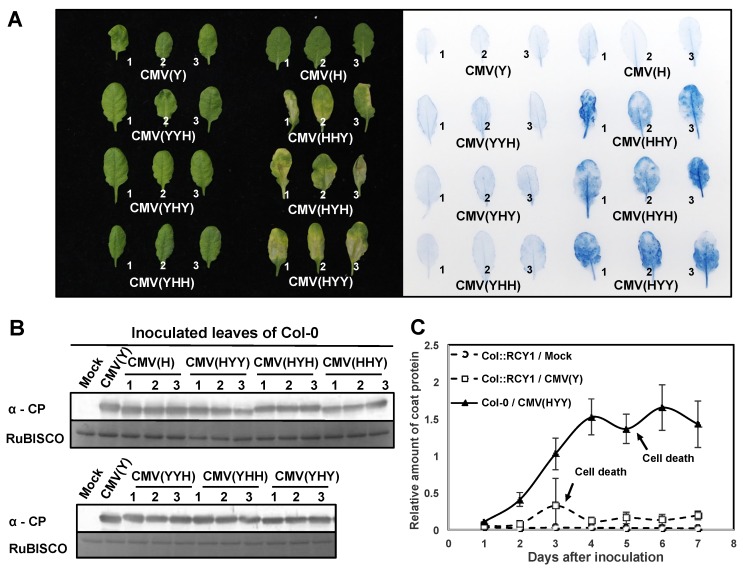
Response of virus-inoculated leaves of *Arabidopsis thaliana* ecotype Col-0 to CMV(H), CMV(Y), or a series of reassortant CMVs, and virus multiplication in the inoculated leaves. (**A**) Development of cell death in leaves with a series of reassortant CMVs, or with CMV(H) or CMV(Y) as a control. Representative virus-inoculated Col-0 of three independent plants (plant numbers 1, 2, and 3) under bright field (left panel) and stained with trypan blue (right panel). (**B**) CMV CP detected immunologically by western blotting at 7 dpi in the leaves of plants inoculated with one of a series of reassortant CMVs. CMV(Y)-inoculated Col-0 leaves and mock-inoculated Col-0 leaves were used as positive and negative control. RuBISCO protein is shown as an internal reference for protein quantity. (**C**) Time course of virus multiplication in Col-0 leaves inoculated with CMV(HYY) carrying CMV(H) RNA1 [Col-0/CMV(HYY)], CMV(Y)-inoculated Col::RCY1 leaves [Col::RCY1/CMV(Y)], and mock-inoculated Col::RCY1 leaves [Col::RCY1/Mock]. CMV CP quantities were measured using ELISA (mean values of relative amount of CP of three independent biological samples with standard error bars).

**Figure 2 viruses-12-00091-f002:**
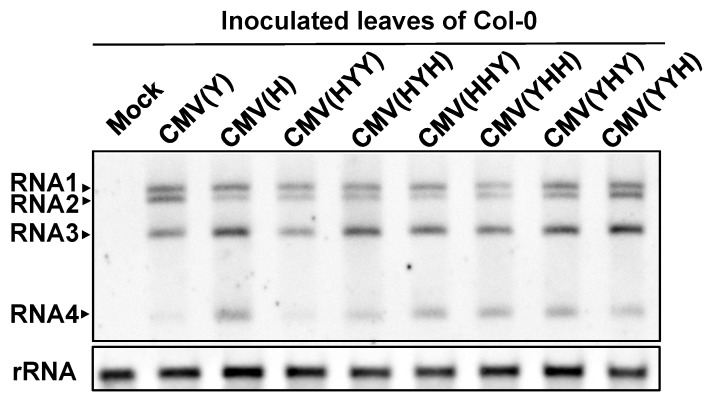
Detection of CMV RNA in virus-inoculated leaves of *Arabidopsis thaliana* ecotype Col-0. CMV RNA1, 2, 3, and 4 were detected by northern blot hybridization analysis of total RNA extracted from virus-inoculated leaves of Col-0 inoculated with CMV(Y), CMV(H) and a series of reassortant CMVs (as indicated) at 5 dpi. Mock-inoculated Col-0 leaves were used as a control. Total RNA was extracted from three independent samples. The position of CMV RNA is indicated at left: RNA 1, 2, and 3 represent genomic RNAs; RNA4 is subgenomic. rRNA is the loading control.

**Figure 3 viruses-12-00091-f003:**
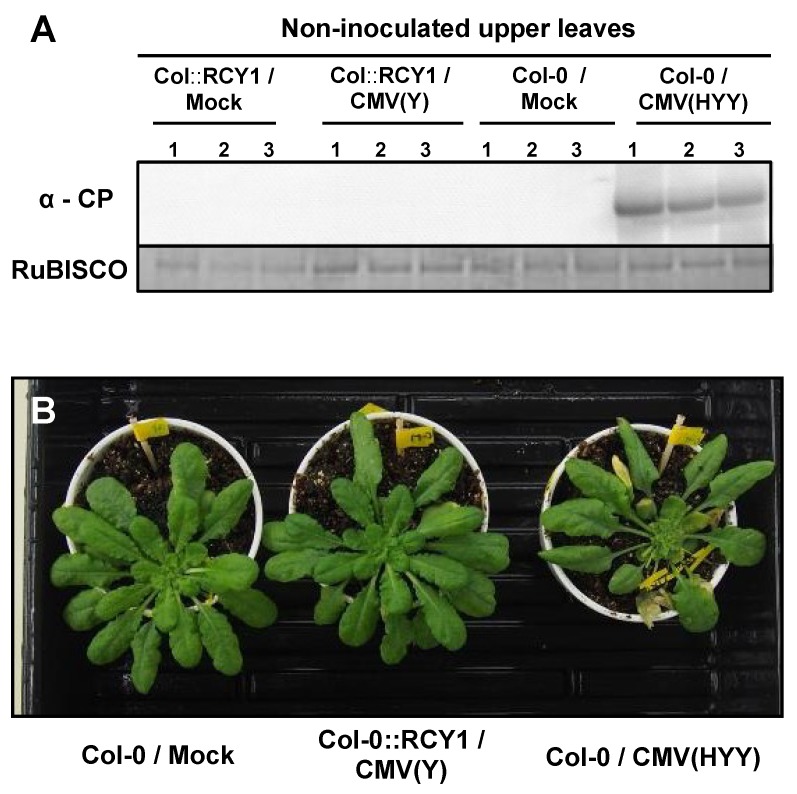
Detection of CMV CP in noninoculated upper *Arabidopsis thaliana* ecotype Col-0 leaves and systemic symptom development. (**A**) CMV CP detected at 7 dpi, by western blot analysis, in noninoculated upper leaves of CMV(HYY)-infected or mock-inoculated Col-0 [Col-0/CMV(HYY) and Col-0/Mock] and CMV(Y)-inoculated or mock-inoculated Col::RCY1 [Col::RCY1/CMV(Y) and Col::RCY1/Mock]. RuBISCO protein is an internal reference for protein quantity. Each experiment was conducted using three independent biological replicates (plant numbers 1, 2, and 3). (**B**) Symptom appearance observed at 14 dpi on CMV(HYY)-inoculated Col-0 [Col-0/CMV(HYY)], CMV(Y)-inoculated Col::RCY1 [Col::RCY1/CMV(Y)], or mock-inoculated Col-0 [Col-0/Mock] (control). Virus-inoculated leaves have been removed because they were already dead at this stage. Representative plants were photographed.

**Figure 4 viruses-12-00091-f004:**
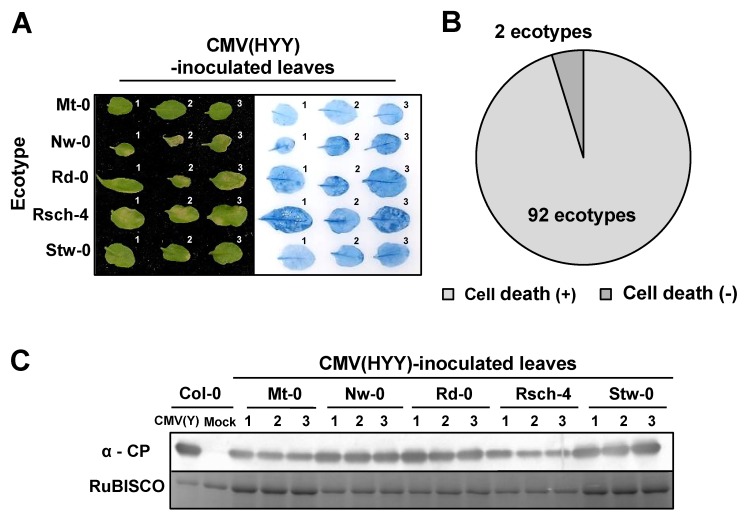
Survey of the response to CMV(HYY)-inoculation on the leaves of 94 ecotypes of *Arabidopsis thaliana*. (**A**) Representative photograph of the responses: CMV(HYY)-inoculated leaves of five ecotypes randomly selected at 14 dpi. Virus-inoculated leaves under bright field (left panel) and stained with trypan blue (right panel). (**B**) Pie chart summary of CMV(HYY)-inoculated leaves of the 94 ecotypes. (**C**) CMV CP detected immunologically by western blot analysis in virus-inoculated leaves of three independent biological replicates (numbers 1, 2, and 3) of five selected ecotypes at 7 dpi. RuBISCO protein is an internal reference for protein quantity.

**Figure 5 viruses-12-00091-f005:**
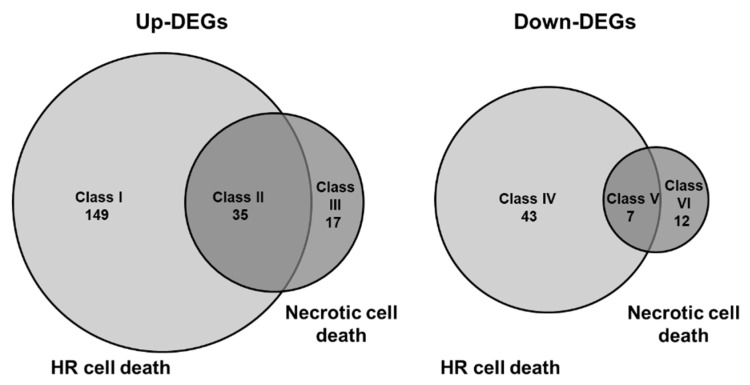
Venn diagram of the number of genes with increased or decreased transcript abundance in CMV(HYY)-inoculated *Arabidopsis thaliana* Col-0 leaves showing necrotic cell death and CMV(Y)-inoculated Col::RCY1 leaves showing HR cell death. The number of genes detected by RNA-Seq analysis with more than 4-fold increased expression and adjusted *p* < 0.05 in CMV(HYY)-inoculated Col-0 leaves and CMV(Y)-inoculated Col::RCY1 leaves are shown at left (Up-DEGs). Those with less than 0.25-fold decreased expression (*p* < 0.05) are shown at right (Down-DEGs). The number of genes with increased and decreased expression in leaves showing HR cell death are shown in the light gray circles; and those showing necrotic cell death in the dark gray circles.

**Figure 6 viruses-12-00091-f006:**
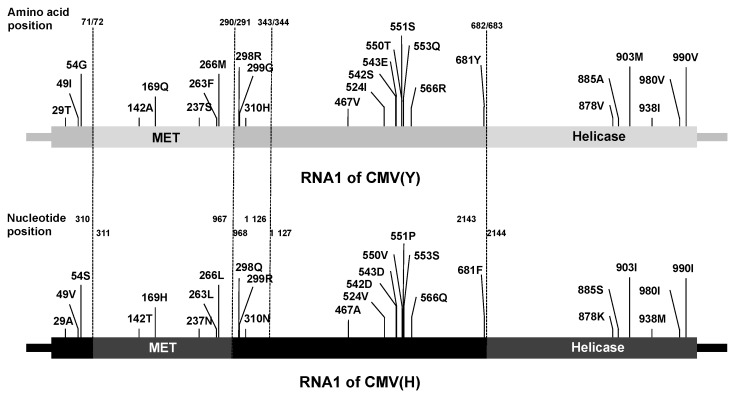
Schematic diagram of the CMV RNA1 encoding the 1a protein, which contains a methyltransferase (MET) and a helicase (HEL) domain. The 1a protein-coding region and corresponding 1a protein are shown as rectangles. Amino acids that differ between the 1a proteins encoded by the CMV(H) RNA1 (lower panel) and the CMV(Y) RNA1 (upper panel) and the positions of these amino acids in the 1a protein are described above each rectangle. The dotted lines connecting the two chimeric constructs indicate the amino acid and nucleotide positions of junction sites. The adjacent sequence numbers of each of these junctions are indicated above the CMV(Y) RNA1 (upper) schematic for amino acids; and above the CMV(H) RNA1 (lower) schematic for the corresponding nucleotides.

**Figure 7 viruses-12-00091-f007:**
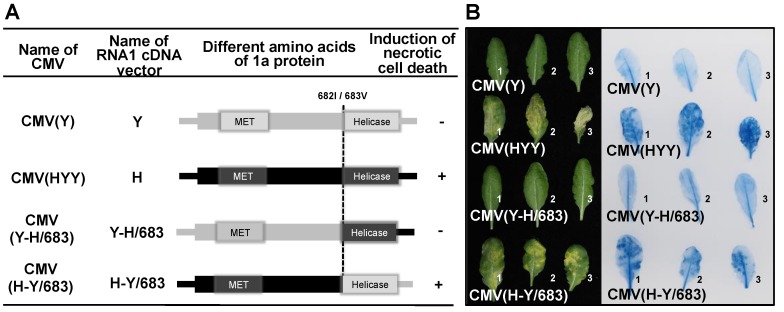
Induction of necrotic cell death in *Arabidopsis thaliana* ecotype Col-0 leaves inoculated with reassortant CMVs, CMV(H-Y/683) and CMV(Y-H/683), which carry chimeric 1a protein of CMV(Y) and CMV(H), CMV(Y), and CMV(HYY). (**A**) Schematic diagram of the CMV RNA1 encoding the 1a protein. The 1a protein-coding region (and its corresponding 1a protein) is presented as rectangles in black for CMV(H) and in gray for CMV(Y). The dotted line indicates the junction site at amino acid position 682/683. MET, 1a protein methyltransferase domain; HEL, helicase domain. The presence (+) or absence (−) of necrotic cell death induction in virus-inoculated leaves is shown in the column on the right. (**B**) Responses of CMV-inoculated leaves for three independent biological replicates (plants number 1, 2, and 3). Virus-inoculated leaves to the left are under bright field and those to the right are stained with trypan blue.

**Figure 8 viruses-12-00091-f008:**
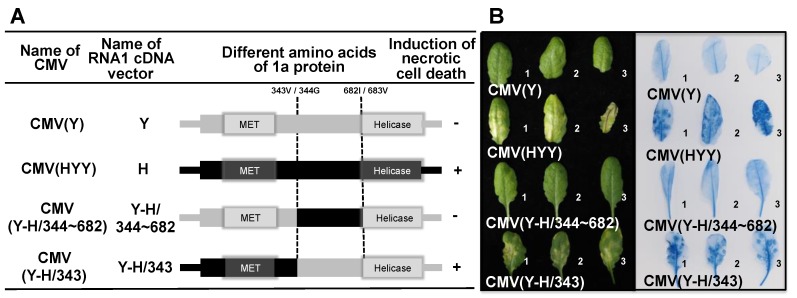
Induction of necrotic cell death in *Arabidopsis thaliana* ecotype Col-0 leaves inoculated with reassortant CMVs, CMV(Y-H/343), and CMV(Y-H/344~682) which carry chimeric 1a protein of CMV(Y) and CMV(H), and CMV(Y) and CMV(HYY). (**A**) Schematic diagram of the CMV RNA1 encoding the 1a protein (representation as in Figure 7: see legend for details). The junction sites in chimeric 1a proteins at amino acid positions 343/344 and 682/683 are indicated by dotted lines. (**B**) Responses of CMV-inoculated leaves (for details, see legend to Figure 7).

**Figure 9 viruses-12-00091-f009:**
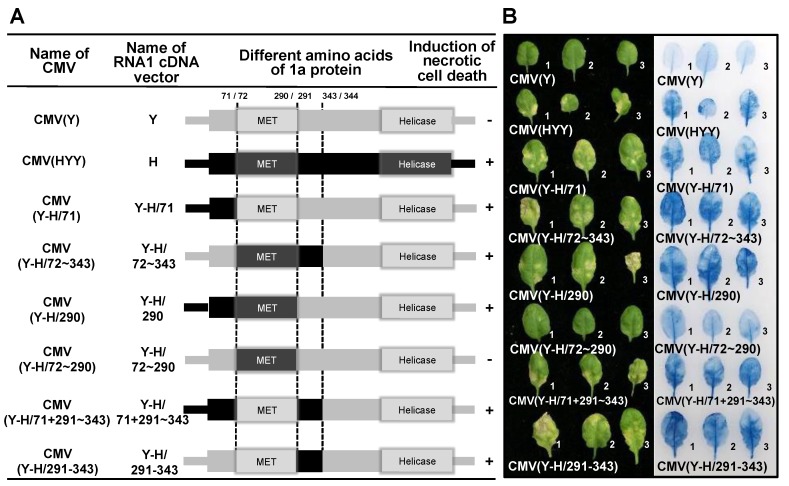
Induction of necrotic cell death in *Arabidopsis thaliana* ecotype Col-0 leaves inoculated with reassortant CMVs, CMV(Y-H/71), CMV(Y-H/72~343), CMV(Y-H/290), CMV(Y-H/72~290), CMV(Y-H/71 + 291~343), and CMV(Y-H/291~343), which carry chimeric 1a protein of CMV(Y) and CMV(H), and CMV(Y) and CMV(HYY). (**A**) Schematic diagram of the CMV RNA1 encoding the 1a protein (representation as in Figure 7: see legend for details). Dotted lines indicate the chimera junction sites at amino acid positions 71/72, 290/291, and 343/344. (**B**) Responses of CMV-inoculated leaves (for details, see legend to Figure 7).

**Figure 10 viruses-12-00091-f010:**
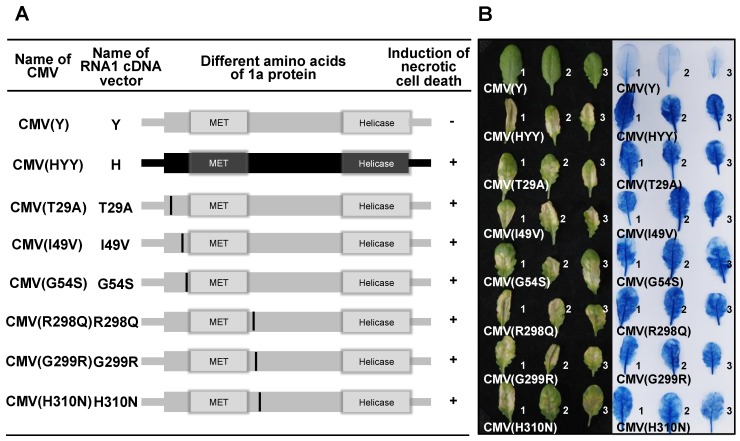
Induction of necrotic cell death in *Arabidopsis thaliana* ecotype Col-0 leaves inoculated with CMVs carrying single amino acid substitutions in the 1a protein of CMV(Y). (**A**) Schematic diagram of the CMV RNA1 encoding the 1a protein. The 1a protein-coding region and corresponding 1a protein are shown as rectangles. Deduced single amino acid substitutions and their positions in the 1a protein are shown as a black bar (representation as in Figure 7: see legend for details). (**B**) Responses of CMV-inoculated leaves (for details, see legend to Figure 7).

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
