# Peer review of "Single Amino Acid Substitutions in the Cucumber Mosaic Virus 1a Protein Induce Necrotic Cell Death in Virus-Inoculated Leaves without Affecting Virus Multiplication"

_viruses, 2020, doi:10.3390/v12010091_

Round 1
Reviewer 1 Report
The authors used a series of CMV reassortants and noticed a peculiar behaviour for some of them in Arabidopsis Col-0, due to appearance of necrotic cell death in the inoculated leaves, with no changes in virus titer nor ability of long distance movement to upper leaves. By mutagenesis they were able to associate the necrosis to aminoacids in the genomic region surrounding the MET domain of the 1a protein. They also performed RNAseq to compared the necrotic response and the HR observed for a strain inducing classical HR in Arabidopsis, and noticed differences in the transcript profiles, supporting the hypothesis that the two observed necrotic responses are the result of different processes, with the necrotic cell death likely do not contributing to resistance to CMV. Overall the manuscript can be followed, even if editing/justification of some figures and statements should be provided. Beside editing suggestions, I have written in the comments below a couple of questions related to the experiments already performed, that might support further investigation or discussion by the authors.
Title: „might confer to induce necrotic cell death“: I think such a title is not very clear, because the statement is confusing and it communicates a possibility rather than a statement. Based on authors statements, the substitutions are associated to necrotic cell death and this should be clearly stated in the title. Further, statement "not affect virus multiplication": please see related comment below.
The last sentence of the abstract sounds speculative and not appropriate in this context.
L20: “did not affect virus multiplication”: in the necrotic inoculated leaves many cells are actually dead. Indeed, if the virus titer does not change, the virus has to replicate more in the remaining living cells, to get the same titer in non-necrotic leaves. Indeed, virus multiplication might occur in a different way in the two conditions. Can the authors elaborate on this?
L29: “the role of cell death”: maybe the necrotic response is a side effect with no role in the interaction… I would be more careful with this statement
L50: “the” necrotic local lesions…
L76: “cell death was identified”
L83: “in Table”
L83: some details on the RCY1-transformed Col-0 would be useful
L212: details quantitative data on the number of leaves used should be given
L275: the text refers to 5 dpi, the Figure to 8 dpi
L276: are the authors still referring to 5 dpi (as in line 274) or is here 8dpi correct?
L282: “were quite similar”: please use a better term for quantification
L285: “which showed a HR response”
L287:” which was susceptible to infection”
L310-311: mentioning only the constructs name would be enough
L328: looking to Fig. 3A, it appears that Stw-0 is actually showing necrosis: leaf 2 shows a clear brown area and indeed the blue is quite intense compared to MT-0. This image is not convincing to me. Further, the necrosis can also appear at different times for different genotypes, as they can respond very differently. Overall, the results described in section 3.2 appear to me not so related to the main topic of the paper, and additional experiments with all the lines should be performed to get better information on the behaviour of each genotype.
L364-380: getting the information on the GO form the Supplementary table is very difficult: I think the authors should provide a better way to summarize the GO analysis. There is also the possibility to do “enrichment analysis” of plant GO terms, and verify which GOs are enriched in a conditions compared to others. Please check the literature for such approach.
L385: the sentence should be revised
Fig. 6: also the images of trypan-treated leaves are in bright field. Please modify all figures.
L394-395: from “containing” to “YI”: not necessary to repeat.
L400: 1a protein of strain Ho
L400: at this point is not defined if this regions “determines the necrotic cell death”. The author can state that “ is necessary for necrotic cell death”.
Fig.7B: the images on the left for strain Y and HYY are clearly the same as in Fig. 6, and in Fig. 9 also, but not in Fig. 8 . Why the blue image is instead different and not corresponding? There is apparently some incongruence here. Please check carefully all details.
L409: indicate nt position in the figure
L418: substitution induced mutations induced?? The term is not correct
L438-441: the sentence is repeating what already stated before.
Fig. 9: Why the trypan blue leaves of HYY look so different compared to former figures? The necrosis is not so evident here. Further: for mutant Y29A (or T29A?, not consistent; please check this in text and figure too), trypan blue staining of necrotic areas is not so evident from the figure and not comparable to the other positive ones. It appears also more in dots rather than as necrotic areas: can the authors clarify this?
L463: “more substantially”: specify better
L471: “which was thought to be symptoms”, needs editing
L471: “may be resulted”, needs editing
One of the major questions left in the manuscript is the contribution of the RNA2/3 to the necrosis. Could the author test if RNA1 alone can induce necrosis? Or could they mix the mutagenized RNA1 with the other combinations (HHY and HYH) and get information from this?
Author Response
Responses to the comments of Reviewer #1
Title: „might confer to induce necrotic cell death“: I think such a title is not very clear, because the statement is confusing and it communicates a possibility rather than a statement. Based on authors statements, the substitutions are associated to necrotic cell death and this should be clearly stated in the title.
Response: Considering the Reviewer’s suggestion, we have changed the title into “Single Amino Acid Substitutions in the N-terminal Region Surrounding the Methyltransferase Domain of the Cucumber Mosaic Virus 1a Protein Could Confer to Induce Necrotic Cell Death in Virus-inoculated Leaves but not Affect Virus Multiplication in Arabidopsis thaliana”.
The last sentence of the abstract sounds speculative and not appropriate in this context.
Response: We have re-written this part according to the Reviewer’s suggestion.
L20: “did not affect virus multiplication”: in the necrotic inoculated leaves many cells are actually dead. Indeed, if the virus titer does not change, the virus has to replicate more in the remaining living cells, to get the same titer in non-necrotic leaves. Indeed, virus multiplication might occur in a different way in the two conditions. Can the authors elaborate on this?
Response:It is true that many cells are dead in the necrotic inoculated leaves. And those dead wizened cells will not be counted into the biomass of samples. Therefore, we suppose that the necrotic inoculated leaves share same concentration of protein in the extraction buffer with other samples. And the process of virus multiplication in the two conditions is indeed interesting, however in this section, we focus on the result of virus multiplication rather than the process of it.
L29: “the role of cell death”: maybe the necrotic response is a side effect with no role in the interaction… I would be more careful with this statement
Response:We agree with you at this point. Indeed, it is still unclear that whether the necrotic cell death is required or just a consequence of the interaction. We are sorry for this confusing statement. Thus, we have re-written this sentence.
L50: “the” necrotic local lesions…
Response:The grammar has been revised according to Reviewer’s suggestion.
L76: “cell death was identified”
Response:The term “cell death was analyzed” has been revised to “cell death was identified” according to Reviewer’s suggestion.
L83: “in Table”
Response:The term “on Table” has been revised to “in Table” according to Reviewer’s suggestion.
L83: some details on the RCY1-transformed Col-0 would be useful
Response:Following the reviewer’s suggestion, we have added some details on the RCY1-transformed Col-0.
L212: details quantitative data on the number of leaves used should be given
Response:According the Reviewer’s suggestion, the number of leaves used in ELISA has been added into the section2.6.
L275: the text refers to 5 dpi, the Figure to 8 dpi
Response:We are very sorry for our negligence of miswriting. The number of dpi should be 5.
L276: are the authors still referring to 5 dpi (as in line 274) or is here 8dpi correct?
Response:We are very sorry for our negligence of miswriting. The number of dpi should be 5.
L282: “were quite similar”: please use a better term for quantification
Response: As suggested, we have revised this term.
L285: “which showed a HR response”
Response:We have changed the sentence according to the Reviewer’s suggestion.
L287:” which was susceptible to infection”
Response:We have changed the sentence according to the Reviewer’s suggestion.
L310-311: mentioning only the constructs name would be enough
Response:We have changed the sentence according to the Reviewer’s suggestion.
L328: looking to Fig. 3A, it appears that Stw-0 is showing necrosis: leaf 2 shows a clear brown area and indeed the blue is quite intense compared to MT-0. This image is not convincing to me. Further, the necrosis can also appear at different times for different genotypes, as they can respond very differently. Overall, the results described in section 3.2 appear to me not so related to the main topic of the paper, and additional experiments with all the lines should be performed to get better information on the behavior of each genotype.
Response:We are grateful for the suggestion. According to our repetitive experiments, we could not detected necrosis induction in CMV(HYY)-inoculated leaves of Stw-0 and Mt-0, even if we cultivated them more than one month after inoculation. And we think that the brown area showing at the tip of leaf 2 is the senescence of tissues but not the necrosis. Please look at attached a photo in which you could only find senescence (necrosis) at the edge of leaves.
It is true that those the timing of necrosis induction in response to CMV(HYY) was different among 92 ecotypes between a range of 5 dpi to 9 dpi, but at 14 dpi, the necrosis developed in all CMV(HYY)-inoculated leaves of 92 ecotypes. Thus, in revised version, we described that “Cell death developed in virus-inoculated leaves of 92 of these ecotypes from 5 dpi to 9 dpi, but not in the ecotypes Mt-0 and Stw-0 at 14 dpi (Figures 3A and B, and Table S4)”.
L364-380: getting the information on the GO form the Supplementary table is very difficult: I think the authors should provide a better way to summarize the GO analysis. There is also the possibility to do “enrichment analysis” of plant GO terms and verify which GOs are enriched in a conditions compared to others. Please check the literature for such approach.
Response:Thank you for this good comment. We have performed a GO enrichment analysis of DEGs in each Class as you suggested. The GO enrichment results in biology process of Class I, Class II, and Class III were listed in Table S11-S13 and the top 10 GO enrichment term in BP were plotted in Figure S8-10. There was no enrichment in Class IV, V and Class VI. And the corresponding method and result have been added.
L385: the sentence should be revised
Response:We are very sorry for the incorrect writing. And it has been revised as you suggested.
Fig. 6: also the images of trypan-treated leaves are in bright field. Please modify all figures.
Response:Indeed, both leaves with and without trypan blue staining are pictured in bright field. Thus, we have modified all figures with this problem. Thank you for your suggestion.
L394-395: from “containing” to “YI”: not necessary to repeat.
Response:Thank you for your suggestion, we have removed this part.
21.L400: 1a protein of strain Ho
Response: As you suggested, we have revised the term ”1a protein” into “1a protein of strain Ho”.
L400: at this point is not defined if these regions “determines the necrotic cell death”. The author can state that “is necessary for necrotic cell death”.
Response:Considering the Reviewer’s suggestion, we have revised the term” determines the necrosis cell death” to “is necessary for necrotic cell death”.
23.Fig.7B: the images on the left for strain Y and HYY are clearly the same as in Fig. 6, and in Fig. 9 also, but not in Fig. 8. Why the blue image is instead different and not corresponding? There is apparently some incongruence here. Please check carefully all details.
Response:The experiments of Fig. 6 and 9 were conducted at the same time for receiving a better-quality result. The leaves for trypan blue staining were sampled and pictured at the same time (same control leaves with different test leaves). After decolorization of trypan blue stained leaves, they were pictured in the 70% ethanol solution. The posture of control leaves was changed by the fluidity of ethanol solution. We are very sorry for making you confused about it. We have added this statement of how the trypan blue staining leaves were pictured into the method section2.7.
L409: indicate nt position in the figure
Response:Thank you for your good comment. Indeed, it is better to indicate the nucleotide position in the figure 5 for easier understanding of the paper. Thus, we have modified the figure 5 according to Reviewer’s suggestion.
L418: substitution induced mutations induced?? The term is not correct
Response:We are very sorry for our incorrect writing. This term has been revised in the manuscript.
L438-441: the sentence is repeating what already stated before.
Response:We have made correction according to the Reviewer’s comments.
Fig. 9: Why the trypan blue leaves of HYY look so different compared to former figures? The necrosis is not so evident here. Further: for mutant Y29A (or T29A? not consistent; please check this in text and figure too), trypan blue staining of necrotic areas is not so evident from the figure and not comparable to the other positive ones. It appears also more in dots rather than as necrotic areas: can the authors clarify this?
Response:Thank you for your comments. The trypan blue staining of leaves in Figure 9 didn’t go very well compared to the former ones. However, the leaves before staining exhibited necrosis obviously in Figure 9. We thought it is possible to address the result that the necrotic cell death was induced in Figure 9 combining the picture of leaves without trypan blue staining.
L463: “more substantially”: specify better
Response:We have made specific statement about the result in RNA-seq for getting the “more substantially” result.
Previously:Furthermore, comparative RNA-Seq analysis of CMV(HYY)-inoculated Col-0 leaves and CMV(Y)-inoculated Col::pRCY1-HA#12 leaves, showed that gene expression in CMV(Y)-inoculated Col::pRCY1-HA#12 exhibiting HR cell death, changed more substantially than did that in CMV(YHH)-inoculated Col-0 leaves exhibiting necrotic cell death.
Currently:Furthermore, comparative RNA-Seq analysis of CMV(HYY)-inoculated Col-0 leaves and CMV(Y)-inoculated Col::pRCY1-HA#12 leaves, showed that over 8 folds and 3.5 folds of DEGs were specifically up and down regulated in CMV(Y)-inoculated Col::pRCY1-HA#12 exhibiting HR cell death compared to CMV(HYY)-inoculated Col-0 exhibiting necrotic cell death, suggesting that gene expression in CMV(Y)-inoculated Col::pRCY1-HA#12 changed more substantially than did that in CMV(HYY)-inoculated Col-0.
L471: “which was thought to be symptoms”, needs editing
Response:We are very sorry for this in correct writing and it has been revised to “which was considered as symptoms”.
L471: “may be resulted”, needs editing
Response:Thank you for your comments. We have discussed it carefully. However, we didn’t see any grammar mistakes in it. I’m afraid there is no need to edit this term.
One of the major questions left in the manuscript is the contribution of the RNA2/3 to the necrosis. Could the author test if RNA1 alone can induce necrosis? Or could they mix the mutagenized RNA1 with the other combinations (HHY and HYH) and get information from this?
Response:You have raised an important point of this study. The necrotic cell death was induced in Col-0 leaves by the infection of reassortant CMVs [CMV(HHY), CMV(HYY), CMV(HYH)]. In brief, for necrotic cell death induction, the RNA1 of CMV(Ho) was required firstly, then the RNA2 and RNA3 of CMV(Y) were optional choices. Combining the comment of Reviewer#2, according to Fig 1A CMV(HYY) and CMV(HHY) show similarly strong staining with trypan blue, while CMV(HYH) has lighter staining more similar to the parental strain CMV(Y) demonstrating that presumable RNA3 has a stronger effect on symptom formation rather than RNA2 of CMV(Y) in the case of the reassortant virus. It is possible to analyze the contribution of RNA2 and RNA3 of CMV(Y) combining with RNA1 of CMV(Ho) in necrotic cell death induction through making constructs RNA2 and RNA3 of CMV(Y) and CMV(Ho) in CMV(HYH) and CMV(HHY), respectively. Since the parental CMV(HHH) lost the ability of induction of necrotic cell death, we suspect that the RNA1 of CMV(Ho) alone could not induce necrotic cell death. We would like to analyze this issue in the next step of our research project.
Reviewer 2 Report
The manuscript by Tian et al. entitled "Single amino acid substitutions in the N-terminal region surrounding the methyltransferase domain of the cucumber mosaic virus 1a protein might confer to induce necrotic cell death in virus inoculated leaves but not affect virus multiplication in Arabidopsis thaliana" analyze the necrosis induction on the inoculated leaves of Arabidopsis thatiana ecotype Col-0 induced by a reassortant CMV. The research topic is interesting since the knowledge of necrosis induction of different viruses is still incomplete. The authors analyze both the viral and plant factors connected to this phenomenon. Even if several experiments are presented, several questions are still pending.
Since the reassortant virus (HYY) has different phenotype than the parental strains (CMV-Y; CMV-Ho), the ratio of the genomic RNAs (RNA1, RNA2, RNA3, RNA4) can be interesting and crucial, so Northern analysis is essential besides the detection of the coat protein by western blot. According to Fig 1A CMV(HYY) and CMV(HHY) show similarly strong staining with tryptan blue, while CMV(HYH) has lighter staining more similar to the parental strain CMV-Y demonstrating that presumable RNA3 has a stronger effect on symptom formation in the case of the reassortant virus. I suggest including also the photo of the infected leaves not just the visualization of cell death.
According to the results, the HYY reassortant has different symptoms on the systemically infected leaves, namely necrotic symptoms are not present. The infection phenotype is similar in the case of all the recombinants and point mutants inducing necrosis on the inoculated leaves. I suggest inoculating Arabidopsis thatiana plants with the sap of the systemically infected leaves to verify the stability of the virus strains. If the mutants are stable, the necrosis will develop on the inoculated leaves again. Sometimes second site mutations can also have an effect on symptom formation.
Minor points:
- The name of the recombinants is confusing. I suggest rename all of the recombinants according to the origin of the different regions of RNA1 and the position of the recombination (for example instead of Y-I use Y-H/683)
- It is confusing that the Fig 6-9 are in the "Materials and Methods" section. Please transfer them to the result section. If necessary, all of the constructs could be outline as a supplementary figure. Fig 6-7 should include in one, especially since the photo of the control CMV(HYY) is identical.
-In Table S3 the bold characters are missing.
-I suggest to include a photo of the necrotic lesion of CMV-Y on the Col-0::pRCY1-HA 12 plants.
Author Response
Responses to the comments of Reviewer #2
Since the reassortant virus (HYY) has different phenotype than the parental strains (CMV-Y; CMV-Ho), the ratio of the genomic RNAs (RNA1, RNA2, RNA3, RNA4) can be interesting and crucial, so Northern analysis is essential besides the detection of the coat protein by western blot.
Response:Thank you for your thoughtful comment. Indeed, we did not conduct northern analysis and will have the plan to do it for next step in our research project, in this revised version, please allow us to discuss that the ratio of the genomic RNAs (RNA1, RNA2, RNA3, RNA4) may contribute to the different phenotype in HYY and parental strain-inoculated Col-0 leaves, and should be analyzed in the next step of our research project.
According to Fig 1A CMV(HYY) and CMV(HHY) show similarly strong staining with trypan blue, while CMV(HYH) has lighter staining more similar to the parental strain CMV-Y demonstrating that presumable RNA3 has a stronger effect on symptom formation in the case of the reassortant virus. I suggest including also the photo of the infected leaves not just the visualization of cell death.
Response:Thank you for your valuable comment. It is interesting to analyze the contribution of RNA2 and RNA3 of CMV(Y) in necrotic cell death induction. And according to Reviewer’s comment, the analysis could be started from producing reassortant CMVs between CMV(HHY) and CMV(HHH). we’d like to further analyze this issue in the next study. And according to the suggestion of Reviewer, we have added the photo of the infected plants to Fig 1.
According to the results, the HYY reassortant has different symptoms on the systemically infected leaves, namely necrotic symptoms are not present.
Response:The compatible interaction between virus-host plant results in systemically infection and necrosis induction. The necrotic cell death could induce systemically resulting plant dying. We agree that the phenotype induced by CMV(HYY) differs from the traditional necrosis. However, according to the results, CMV(HYY)-Col-0 is considered as a compatible interaction. Thus, we prefer to describe the cell death induced in this interaction as necrotic cell death.
The infection phenotype is similar in the case of all the recombinants and point mutants inducing necrosis on the inoculated leaves. I suggest inoculating Arabidopsis thaliana plants with the sap of the systemically infected leaves to verify the stability of the virus strains. If the mutants are stable, the necrosis will develop on the inoculated leaves again. Sometimes second site mutations can also have an effect on symptom formation.
Response:Thank you for your comments here. We have clarified that the virus multiplication in the upper leaves was same to others. Further, we also extracted the total RNA of the virus from the upper leaves and confirmed the sequence of RNA1 without changes. Therefore, we believe that the viruses are stable in the plant.
The name of the recombinants is confusing. I suggest rename all of the recombinants according to the origin of the different regions of RNA1 and the position of the recombination (for example instead of Y-I use Y-H/683)
Response:We are very sorry for the confusing naming. We have revised all the names according to Reviewer’s suggestion.
It is confusing that the Fig 6-9 are in the "Materials and Methods" section. Please transfer them to the result section. If necessary, all of the constructs could be outline as a supplementary figure. Fig 6-7 should include in one, especially since the photo of the control CMV(HYY) is identical.
Response:We are very sorry for the formation of the figures in this manuscript. We have modified the figures according to Reviewer’s suggestion.
In Table S3 the bold characters are missing.
Response:We are sorry for our negligence of formation in Table S3. We have revised it following Reviewer’s comment.
I suggest to include a photo of the necrotic lesion of CMV-Y on the Col-0::pRCY1-HA 12plants.
Response:Thank you for the suggestion. We added the photo of HR cell death in Col-0::pRCY1-HA#12 inoculated with CMV(Y) to Figure S2 as a panel A, and stated it in the text.
Round 2
Reviewer 1 Report
Comments on revised manuscript:
Title: The title remains to me not satisfactory. The authors continue to use an hypothetical statement (“could confer…”)
With reference to the file with “no track changes”
L20-“virus multiplication”: the response to the comments is that the authors “suppose that the… leaves share the same concentration of protein”: indeed, this is a supposition. No data are present on virus multiplication (statistical analysis, RealTime PCR for example). What the authors can say is that “ cell death did not affect the ability of the virus to multiply..”
L76-77: the sentence added on the transformed plant needs English revision
L213: The legend still reports 8dpi; the clarification asked has not been finally performed
L240: the authors mention in the text 14 dpi, but in the response they state that the plants were observed for more than one month. This information should be indeed part of the manuscript.
L244: on the inoculated leaves
Fig.9: the authors state that in the comments the “ the staining …didn´t go very well”. The authors should indeed clarify that the trypan blue staining has limitation in some cases This could be a useful information. A justification on the inefficient staining for HYY should be given, as this appears very evident.
The comments to Reviewers 2 concerns have been mostly addressed
regarding figure formatting or by providing additional figures of
symptomatic plants. In relation to the 2 major substantial points, the
authors ask to postpone these experiments to a future work. As they are
indeed a request of the Reviewer and they are related to this work, it
would be better to see the results in this context. Especially
experiment 1 (first comment) is defined as essential by the Reviewer 2,
but not performed by the authors in the revision. For experiment 2
(second comment), the authors propose to generate new reassortants, but
actually the constructs are already available; it´s not clear the
necessity to make more reassortants. And further, in a bigger view, it
is true that CMV-HYH shows a lighter blue color in Figure 1, but that is
one image only! Was this result really consistent in the different
experimental replicates, so to be able to generalize such an observation?
Author Response
Here we submit our revised manuscript (ID: viruses-621629).
We have extensively revised the manuscript according to your suggestions, also enlisting the help of an external advisor who is a professional biologist and native English speaker. We have marked all the revisions in the manuscript using the Track Changes function in Microsoft Word for your convenience. Our point-by-point revisions in the manuscript and responses to each of the comments of the reviewers are provided in the following pages.
I have uploaded our revised manuscript including figures, supplemental figures and tables, and our response to reviewers’ comments. One further original data item (a northern blot) has been newly added to this revised manuscript, which also has been uploaded.
We would be most grateful if the revised manuscript could be further reviewed and considered for publication in “Viruses”.
As I mentioned before, publication of this manuscript by the middle of January is very important for the first author, who is a PhD candidate hoping to go forward to his Final PhDDegree Screening shortly. Your help and that of the reviewers in expediting the processing of this MS and decision on publication would, therefore, be greatly and most humbly appreciated.
With thanks for your attention and time
Responses to the comments of Reviewer #1
Thank you for your review of our manuscript. We much appreciate your valuable comments and suggestions. Below are our responses to each of the comments.
1) Title: The title remains to me not satisfactory. The authors continue to use an hypothetical statement (“could confer…”)
Response:We have revised the title in line with this suggestion as: “Single Amino Acid Substitutions in the Cucumber Mosaic Virus 1a Protein Induce Necrotic Cell Death in Virus-inoculated Leaves without Affecting Virus Multiplication ”.
2) With reference to the file with “no track changes”
Response:We apologize for losing the "track changes" function in the References section. This has now been rectified.
3) L20 “virus multiplication”: the response to the comments is that the authors “suppose that the… leaves share the same concentration of protein”: indeed, this is a supposition. No data are present on virus multiplication (statistical analysis, Real-time PCR for example). What the authors can say is that “cell death did not affect the ability of the virus to multiply.”
Response:In our experiment, we estimated virus multiplication through the accumulated level of virus coat protein, since coat protein accumulation is generally correlated with virus multiplication. In order to provide a more precise measure, we conducted northern blot analysis to detect CMV RNA in the leaves inoculated with CMV(H), CMV(Y), CMV(HYY), CMV(HYH), CMV(HHY), CMV(YHH), CMV(YHY) and CMV(YYH). As can be seen upon inspection of Figure 2, there is no difference in the band intensities among CMV(HYY), CMV(HYH) or CMV(HHY)-inoculated leaves showing necrotic cell death and CMV(H), CMV(Y), CMV(YHH), CMV(YHY) or CMV(YYH)-inoculated leaves showing no necrotic cell death. We propose that this represents a reasonable compromise.
4) L76-77: the sentence added on the transformed plant needs English revision
Response:This sentence has now been revised accordingly.
5) L213: The legend still reports 8dpi; the clarification asked has not been finally performed
Response:I apologize for this error, which has now been rectified.
6) L240: the authors mention in the text 14 dpi, but in the response they state that the plants were observed for more than one month. This information should be indeed part of the manuscript.
Response:To address this issue, we have added the statement “In repetitions of these experiments, necrosis induction was never detected in CMV(HYY)-inoculated leaves of Stw-0 and Mt-0, even if they were cultivated for more than one month after inoculation(data not shown).”.
7) L244: on the inoculated leaves
Response: As suggested, we have revised this term.
8) Fig.9: the authors state that in the comments the “the staining …didn´t go very well”. The authors should indeed clarify that the trypan blue staining has limitation in some cases This could be a useful information. A justification on the inefficient staining for HYY should be given, as this appears very evident.
Response:
We have added the following statement to section 2.5: “Trypan blue staining is available to detect cell death qualitatively, but it has limitations in attempting to show a quantitative measure of cell death.”.
In addition, we repeated experiments with trypan blue staining to further confirm whether or not cell death was induced in each set of experiments. Representative photographs are presented in revised manuscript.